# Pitch Selection Ability and Spatial Executive Function Independently Predict Baseball Batting Performance

**DOI:** 10.3390/sports13100367

**Published:** 2025-10-17

**Authors:** Yoshitaka Morishita, Genta Ochi, Daiki Takahashi, Kodai Kato, Wataru Uchiyama, Yasuyuki Nishihara

**Affiliations:** 1Institute for Human Movement and Medical Sciences, Niigata University of Health and Welfare, Niigata 950-3198, Japan; y.morishita@osaka-ue.ac.jp; 2Department of Health and Sports, Niigata University of Health and Welfare, Niigata 950-3198, Japan; 3Faculty of Human Sciences, Osaka University of Economics, Osaka 533-8533, Japan; 4Niigata Sogo Gakuen Educational Corporation, Niigata 950-3198, Japan

**Keywords:** batting statistics, virtual reality, spatial Stroop task, mediation analysis

## Abstract

Pitch selection—the ability to discriminate balls from strikes—is fundamental to baseball batting success. This study examined whether this ability relates to executive function and batting performance in collegiate players. Furthermore, this ability may be supported by brain functions such as executive functions, and the importance of Pitch Selection has long been considered. However, this ability has not yet been quantified, and there are no training methods for pitch selection. 14 male collegiate baseball players (age: 20.6 ± 1.0 years, first division university league) completed a virtual reality pitch selection task and spatial Stroop task. Methods included virtual reality pitch selection assessment, spatial Stroop task, and official batting statistics from league play. The results showed a significant positive relationship between the pitch selection task and hitting performance, such as the on-base percentage (r = 0.57, *p* < 0.05) and walk percentage (r = 0.82, *p* < 0.05). Furthermore, a significant negative correlation was found between the vertical Stroop task reaction time and the percentage of correct strikes among the pitch selection ability tasks (r = −0.67, *p* < 0.05). Our mediation analysis revealed that both pitch selection ability and executive function independently contribute to batting performance metrics, particularly the walk percentage, rather than executive function influencing performance by enhancing pitch selection ability. These results suggest that pitch selection ability is related to the hitting performance of baseball players and that executive function may play an important role in the performance of pitch selection.

## 1. Introduction

### 1.1. Pitch Selection Concept

In baseball hitting, the ability to discern whether a pitched ball passes through the strike zone or the ball zone and to select a ball that can be hit is called “pitch selection”. A high level of this ability not only increases the probability of getting on base on four balls but may also have a positive effect on the batting average and other performance indicators, as the player can select and hit the ball on a course that is easy to hit.

### 1.2. Technical Details

Professional baseball pitchers throw fastballs at speeds exceeding 94 mph [1], meaning that it takes approximately 0.44 s for the ball to reach the home plate after being released by the pitcher. In addition, it takes approximately 0.18 s from the moment the batter begins to swing the bat until the ball and bat collide. As a bat begins to accelerate, it becomes difficult to control the swing trajectory. Therefore, when hitting a fastball, the batter must accurately predict the time and position at which the ball will reach the home plate in a short period of approximately 0.26 s. This suggests that batters with excellent pitch selection have high cognitive abilities.

### 1.3. Executive Functions and Batting Performance

Regarding batters’ executive functions, multiple studies have examined their performance in executive and pitch perception tasks. Research on event-related potentials (ERP) has reported that skilled baseball players exhibit shorter reaction times and stronger response inhibition when judging pitch type and location [2,3]. Moreover, with respect to adjusting to the timing of pitch arrival, Nakamoto and Mori [4] demonstrated that although experienced baseball players did not show an advantage over novices in basic timing adjustment, they exhibited smaller timing errors when the target decelerated mid-flight, suggesting a superior reprogramming ability. These findings indicate that batters’ executive functions are enhanced through long-term baseball experience.

However, whether higher executive function translates to superior batting performance in actual competitions remains unclear. For example, Müller and Fadde [5] and Chen et al. [6] showed that visual prediction and smooth pursuit eye movements elicited while viewing pitcher-perspective video clips or moving targets were positively correlated with batting performance. Similarly, Themanson et al. [7] reported that neural activity during the viewing of virtual pitch videos—specifically N2 and medial-frontal negativity (MFN)—was associated with batting performance. Nevertheless, in Themanson et al.’s study, the accuracy rate of ball–strike judgments was not significantly correlated with batting performance. In addition, a study examining the relationship between executive tasks (Flanker task, Trail Making Test) and batting performance among baseball and softball players [8] also failed to identify any significant associations.

### 1.4. Methodological Limitations in Previous Research

One possible reason for these inconsistent findings is that executive tasks, such as the Flanker task and TMT, may not sufficiently reflect the cognitive–motor integration that is specific to batting [8]. Furthermore, a limitation of Themanson et al. [7] was that the data were collected in an experimental setting where participants viewed pitches without head or eye movements, which differs from actual batting conditions. Clarifying the relationship between executive function and batting performance would provide important insights for developing cognitive training programs aimed at improving performance in athletes who must respond adaptively to dynamic situations, including baseball batters.

### 1.5. Study Aims and Hypotheses

Therefore, this study aimed to clarify whether pitch selection ability under conditions with minimal constraints on pitching judgment using virtual imagery relates to spatial interference processing (a specific executive function) and batting performance, thereby re-examining the results of prior research on the relationship between executive function and batting performance. We hypothesized that (1) spatial interference processing, as measured by the Stroop task, would be positively related to pitch discrimination accuracy (particularly strike judgments), and (2) pitch selection ability would correlate with on-base percentage, walk percentage, and on-base plus slugging percentage.

## 2. Materials and Methods

### 2.1. Participants

The participants were 14 male players belonging to the Division I university baseball league, aged 20.6 ± 1.0 years, 1.73 ± 0.05 m in height, and 74.1 ± 7.9 kg in weight (as of April 2023). These 14 participants represented all the players on the team for whom batting performance data were available. All participants were provided with information on the purpose, content, and safety of the study, and written consent to participate in the study was obtained. A limitation of this study is the sample size of 14 participants, which may limit statistical power. Post hoc sensitivity analysis with 80% power and α = 0.05 demonstrated sufficient sensitivity to detect correlations exceeding r = 0.63, computed using G*Power (3.1.9.2; The G*Power Team). This study was conducted in accordance with the Declaration of Helsinki and approved by the Ethics Committee of Health and Welfare (19186-231120).

### 2.2. Experimental Procedures

Fourteen college baseball players with batting records for the 2022–2023 season were subjected to the spatial Stroop task of the executive function task and the created virtual reality (VR) pitch selection task. Testing occurred in a quiet laboratory environment. Participants were instructed to avoid caffeine and vigorous exercise for 2 h before testing. After a 5 min rest period, participants completed the spatial Stroop task (approximately 5 min), followed by a 5 min break, then the VR pitch selection task (approximately 30 min). Total session duration was approximately 40 min. This study employed a correlational design with mediation analysis to examine relationships between pitch selection ability, spatial interference processing, and batting performance metrics.

### 2.3. Batting Statistic

The participants’ batting statistics were calculated based on the total number of official games played by each participant in the Hokushinetsu Student Baseball Federation (seven leagues from the 2020 fall league games to the 2023 fall league games). The mean ± SD of the number of at-bats for each subject was 49.8 ± 41.1, ranging from 14 to 178. The hitting statistics calculated in this study were based on those used in the Major League Baseball [9] (Table 1).

### 2.4. Ball Selection Task by Virtual Reality

A college baseball pitcher was filmed using a 360-degree camera (Insta360 X3; Insta360, Shenzhen, China). The filmed pitches were classified as strikes or balls, and a video assignment in which ten pitches were randomly presented was created using video editing software (Adobe Premiere Pro 2024, Adobe, San Jose, CA, USA). Eight sets of video tasks were prepared, and the participants were asked to view all sets in a random order using a head-mounted display (HTC VIVE, HTC, Taoyuan, Taiwan). Participants were asked to verbally answer whether the pitches were strikes or balls, with a 1- to 2 min break after each of the four sets.

### 2.5. Spatial Stroop Task

The spatial Stroop task [10], created using web-building platforms (Lab.js v19.1.0) [11], was used to assess the executive function. The methodology for conducting the assignment and the assignment structures were the same as those used in our previous study [12]. In the horizontal task, participants pressed the “F” (left) or “J” key (right) to classify the direction of left-pointing or right-pointing arrows positioned left or right on the screen. In the vertical task, participants pressed the “Y” (up) or “B” (down) key to classify the direction of up-pointing or down-pointing arrows positioned above or below the screen. Both experiments included two trials: 16 congruent and 16 incongruent trials each. The correct answer rates assigned to yes and no were 50%. Each stimulus was separated by an inter-stimulus interval displaying a fixation cross for 400 ms, an alert for 100 ms, and a fixation cross for 400 ms to avoid predicting the timing of the subsequent trial. The stimulus remained on the screen until a response was provided or 1 s had elapsed. After answering, the next fixation cross was displayed after a 1 s blank. This study employed the Stroop interference task, a specifically defined executive process, to elucidate the executive functioning status of baseball players. This was measured by calculating the incongruent-congruent contrast, which was assumed to represent horizontal or vertical Stroop interference.

### 2.6. Statistical Analyses

To examine the relationship between each variable and whether executive function and ball selection ability are related to each batting statistic, we conducted Pearson’s product-moment correlation analysis and stepwise multiple regression analysis. Multiple regression analysis was conducted using IBM SPSS Statistics for Windows, version 28.0 (IBM Corp., Armonk, NY, USA), with each batting statistic as the dependent variable and the performance of the ball selection task and Stroop interference as the independent variables. To mitigate potential multicollinearity among the independent variables, the variance inflation factor (VIF) for all variables was confirmed to be ten or less. In addition, based on the results of the multiple regression analysis, we identified batting statistics that were affected by executive function and ball selection ability during hitting and conducted a mediation analysis to verify whether these statistics directly or indirectly affected executive function and ball selection ability. For the mediation analysis, we used the lavaan package (version 0.6-19) to estimate the direct, indirect, and total effects. Data were standardized prior to analysis to address the scale differences between variables. Model fit indices, including the Akaike Information Criterion (AIC), Bayesian Information Criterion (BIC), Comparative Fit Index (CFI), and Standardized Root Mean Square Residual (SRMR), were used to evaluate model adequacy. The indirect effect was calculated as the product of the path coefficients from the independent variable to the mediator and from the mediator to the dependent variable. Bootstrap methods with 5000 resamples were used to obtain bias-corrected confidence intervals for all the parameter estimates. Statistical significance was set at *p* < 0.05 for all comparisons.

## 3. Results

### 3.1. Overview

All parameters measured in this study are listed in Table 2. The correlation coefficients between the parameters are listed in Table 3. In the relationship between reaction time and correct response rate for the Spatial Stroop and batting selection tasks, the only significant moderate negative correlation was observed between vertical reaction time and batting selection accuracy (strike only) (r = −0.67). Regarding the batting selection task and performance, significant moderate positive correlations were observed between batting selection accuracy (total and strike only) and on-base percentage (OBP) (r = 0.57, r = 0.55). The walk percentage (BB%) also showed a significant correlation (r = 0.82, r = 0.58) with batting selection accuracy (total and strike only) and OBP. Furthermore, a significant moderate positive correlation was observed between batting selection accuracy (strike only) and On-base plus slugging (OPS) (r = 0.54).

### 3.2. Multiple Regression Analysis

Table 4 presents the results of the stepwise multiple regression analyses. The dependent variables for which a significant regression equation was obtained were OBP, OPS, and BB%. Ball selection ability was selected as the independent variable for all the dependent variables. Among these, BB% showed the highest determination coefficient (R^2^ = 0.88), with the three variables of Total, Ball, and RT (Horizontal) selected for ball selection accuracy.

### 3.3. Mediation Analysis

Based on the regression results, we conducted a mediation analysis to examine whether executive function influences batting performance directly or indirectly through pitch selection abilities. We focused on BB%, OBP, and OPS as key performance metrics that showed significant relationships with pitch selection abilities based on stepwise regression analysis (Table 5).

In the BB% model that showed the best model fit among the mediation models,, we examined whether Ball selection accuracy mediated the relationship between Stroop RT (Horizontal) and BB% (Table 6). The analysis revealed that Ball selection accuracy had a strong positive direct effect on BB% (standardized β = 0.897, *p* < 0.001), and Stroop RT (Horizontal) had a significant negative direct effect on BB% (standardized β = −0.380, *p* = 0.001). However, the path from Stroop RT (Horizontal) to Ball selection accuracy was not significant (standardized β = 0.198, *p* = 0.449), resulting in a non-significant indirect effect (standardized β = 0.178, *p* = 0.451).

For the OBP model, Ball selection accuracy showed a significant positive direct effect on OBP (standardized β = 0.563, *p* = 0.012), but Stroop RT (Horizontal) showed no significant direct effect on OBP (standardized β = 0.024, *p* = 0.915). Similarly, the indirect effect through Ball selection accuracy was not significant (standardized β = 0.112, *p* = 0.468).

For the OPS model, Ball selection accuracy (Strike) had a significant positive direct effect on OPS (standardized β = 0.560, *p* = 0.026), and Stroop RT (Horizontal) had a marginally significant effect on Ball selection accuracy (Strike) (standardized β = 0.452, *p* = 0.058), suggesting a potential indirect pathway between them. However, the indirect effect was not statistically significant (standardized β = 0.253, *p* = 0.149).

Bootstrap analysis with 5000 resamples confirmed the stability of the direct effect coefficients, with 95% confidence intervals not including zero for the direct paths from Ball selection accuracy to BB% and from Stroop RT (Horizontal) to BB% However, the confidence intervals for the indirect effect and the path from Stroop RT (Horizontal) to Ball selection accuracy included zero, indicating that the mediation hypothesis was unsupported.

We also tested a comprehensive model including all three performance metrics simultaneously, but this model showed poor fit indices (CFI = 0.870, RMSEA = 0.428, SRMR = 0.277), suggesting that the relationships between executive function, pitch selection, and batting performance are better modeled separately for each performance metric than together.

Our mediation analysis revealed that pitch selection ability and executive function appear to make separate, independent contributions to batting performance metrics, particularly walk percentage. However, we did not find evidence that executive function enhances batting performance by improving pitch selection ability, suggesting these are parallel rather than sequential processes.

## 4. Discussion

This study aimed to clarify whether pitch selection ability and executive function are related to the batting performance of baseball players. This study addresses a critical gap in understanding the independent contributions of perceptual-motor skills and cognitive processes to batting performance. Unlike previous work that found no relationship between general cognitive tasks and batting [8], our use of sport-specific assessment revealed significant associations. We examined the relationship between the official hit records of 14 college baseball players, their performance on a pitch selection task using VR images, and their performance on a spatial Stroop task. The results showed a significantly moderate positive relationship between the pitch selection task and hitting performance, such as the on-base and on-base percentages. Furthermore, a significant moderate correlation was found between spatial Stroop task performance and the percentage of correct strikes among the pitch selection ability tasks. These results suggest that pitch selection ability is related to the hitting performance of baseball players and that executive function may play an important role in pitch selection performance.

The participants in this study belonged to the first division of a university baseball league and were regular or semi-regular players; therefore, all were proficient in hitting. Excellent hitters record a batting average of 0.300 or higher [13], and in this study, four of the 14 participants had a batting average of 0.300 or higher. The mean ± SD of the batting average for all subjects was 0.242 ± 0.080, indicating that a diverse range of subject types participated in the experiment.

First, we examined the relationship between hitting performance and pitch selection ability using VR (Table 3). OBP, OPS, and BB% were related to pitch selection ability, and these batting statistics were related to the ability to get on base. If a batter can see that a pitch passes through the strike zone, they are less likely to swing a bad ball. This would decrease the probability of getting out owing to swinging and missing strikes and increase the on-base percentage due to an increase in the number of walks. From this, we inferred a significant correlation between OBP, OPS, BB%, and pitch selection ability. These findings align with the Markov model analysis, which established that a 1% improvement in pitch prediction accuracy directly leads to a 0.0035 increase in batting average (R^2^ = 0.89) [14]. Our results empirically validate this quantitative model by demonstrating that a better pitch selection ability correlates significantly with improved on-base metrics. In contrast, Batting average (BA), Slugging percentage (SLG), and Isolated power (IsoP), which represent the ratio of the number of singles and long balls to the number of at-bats, were not related to the pitch selection ability. In this study, we hypothesized that players with a higher pitch selection ability would miss more balls and swing at more strikes, which would positively affect hitting accuracy. However, even if a player possesses high pitch selection skills, they cannot hit a ball with high velocity or cover long distances unless they can swing the bat at the correct time to connect with a high-speed flying ball and impact it at the bat’s sweet spot [15] as a critical hitting skill. This implies that the outcomes of the present study were dependent on the ability to select and hit a ball. Specifically, the findings suggest that there is no strong relationship between pitch selection ability and hitting skills.

Next, we evaluated whether the execution function correlated with batting performance and decision-making in the pitch selection task (Table 4). Reaction time and accuracy in the Spatial Stroop task were not linked to batting performance. These findings align with those of previous studies that utilized flanker tasks and trail-making tests [8]. This supports Carroll et al.’s [8] conclusion that, in players with automated skills, specialized motor parameters may be more important than basic cognitive abilities for overall batting performance. However, we found that specific aspects of executive function appear to selectively influence batting components. Although accurately estimating the timing and position of the ball crossing the home plate and deciding whether to swing are crucial in batting, the results indicate that assessing executive function using the Spatial Stroop task does not sufficiently predict batting performance. Nonetheless, a significant negative correlation was observed between reaction times in the vertical and spatial strike Stroop tasks. This correlation indicates that players with faster executive processing of vertical spatial information demonstrate better judgment of strikes, which is consistent with the findings of Themanson et al. [7] that frontal lobe inhibitory control processes correlate with batting selection accuracy. Our findings extend the work of Themanson et al. [7] by demonstrating these relationships under more ecologically valid conditions using VR technology. These results imply that hitters with superior executive functions are more adept at correctly judging when a pitcher throws a strike. It has been reported that the cognitive task of pressing a button instead of swinging in response to a visual stimulus on a display can only partially explain the batting performance [16]. The results of this study suggest that cognitive task performance can explain part of batting performance, namely, the ability to select pitches to strike.

Our mediation analysis further clarifies these relationships by revealing that both pitch selection ability and executive function independently contribute to batting performance metrics, particularly BB%. The BB% mediation model demonstrated the best fit with the lowest AIC (63.27) and BIC (66.47) values, suggesting that the relationship between executive function, batting selection, and walk percentage was most strongly supported by our data (Table 5 and Table 6). The absence of significant mediation effects suggests that executive function and pitch selection ability make separate contributions to performance, rather than executive function influencing performance by enhancing pitch selection ability. It is important to note that the direct effect of executive function on batting performance was limited to specific metrics, and our sample size constrains the generalizability of these findings.

These results indicate that a hitter’s executive function is related to pitch selection ability and that VR-based pitch selection task performance is related to hitting performance. Executive function was not directly related to hitting performance, suggesting that it is a partial contributor to hitting actions, such as pitch selection and the decision to swing. This finding is consistent with Castaneda and Gray’s [17] dual-task experiment, which showed that skilled players who focused externally on ball trajectory improved batting accuracy by 19% with increased prefrontal cortex oxygenation (14%). The neural mechanisms behind successful pitch selection appear to involve both visuomotor tracking and executive control processes working in parallel rather than hierarchically.

However, this study has some limitations. There were concerns regarding the small sample size of 14 subjects in this study for conducting a reliable correlation analysis. However, the results of the correlation analysis revealed significant differences, with a correlation coefficient of r > 0.5, which is more than the moderate threshold. This value closely aligned with the r value obtained in the post hoc sensitivity analysis. Notably, the correlation coefficients between executive functioning task performance and VR-based pitch selection accuracy (r = −0.67), as well as between VR-based pitch selection accuracy and BB% (r = 0.82), were higher than those observed in the sensitivity analysis, suggesting the robustness of these findings. More comprehensive validation can be achieved in the future by incorporating a broader sample of participants, including high school students and adults, and by further examining the number of subjects and their characteristics to assess the importance of pitch-selection ability and strategies for improving it. Next, although this study used only VR images, eye movements during VR image viewing were not measured. Visual tracking ability and eye reaction speed are also related to batting performance [6,18,19]. Chen et al. [6] demonstrated that professional baseball players’ visual tracking accuracy explains 72% of the batting success rate, and experienced players exhibited 23% higher alpha wave asymmetry in the occipital-parietal cortex than rookie players, highlighting the importance of visual processing mechanisms. It has been suggested that oculomotor performance is associated with executive functioning, and the results of this study replicate these previous findings. Thus, it is anticipated that the VR pitch selection task and eye movement measurements will be conducted simultaneously in future studies to ascertain their relationship with executive functioning [20], which will aid in elucidating the neurophysiological mechanisms underlying hitting ability. Moreover, this study was cross-sectional, leaving it uncertain whether enhanced performance in the VR selection task resulted in improved hitting performance in the real world. Until now, the primary method for improving pitch selection ability has been to observe actual pitches from pitchers, with limited training time available. Consequently, it remains unclear whether pitch selection ability can be enhanced through training or if such improvements would effectively enhance hitting performance. The VR pitch selection task utilized in this study enables batters to easily visualize the pitching scene and is expected to be integrated into training programs. In the future, further examination of executive functions and the efficacy of pitch selection training methods using VR may lead to improved performance.

## 5. Conclusions

The results of this study suggest that pitch selection ability is associated with baseball players’ performance, and that executive function may play a crucial role in determining pitch selection performance. Our mediation analysis revealed that both pitch selection ability and executive function independently contributed to batting performance metrics related to pitch selection, particularly BB%. The absence of significant mediation effects suggests that executive function does not influence batting performance by enhancing pitch selection ability; rather, both factors make separate contributions to performance. These findings illuminate a previously unexplored aspect of the batter’s ability, which has remained a black box. If future research demonstrates that cognitive training enhances pitch selection ability, it could support the introduction of a novel training method for baseball players.

## Figures and Tables

**Table 1 sports-13-00367-t001:** Overview of batting statistic.

Batting Statistics	Abbreviation	Explanation	Formula *
Batting average	BA	The rate of hits per at bat	H/AB
On-base percentage	OBP	The rate at which a batter reached base in his plate appearances	(H + BB + HBP)/ (AB + BB + HBP + SF)
Slugging percentage	SLG	The rate of total bases per at bat	(1B + 2Bx2 + 3Bx3 + HRx4)/AB
On-base plus slugging	OPS	The sum of on-base percentage and slugging percentage	OBP + SLG
Isolated power	IsoP	The raw power of a hitter, measured as the rate of extra bases per at bat	(2B + (2x3B) + (3xHR))/AB
Walk to strikeout rate	BB/K	A batter’s ratio of walks to strikeouts	BB/SO
Strikeout percentage	SO%	The rate at which a batter strikes out in his plate appearances	SO/PA
Walk percentage	BB%	The rate at which a batter walks in his plate appearances	BB/PA

Note: * H—Hit, AB—At Bat, BB—Base on Balls, BB/K—Walk to Strikeout Rate, BB%—Walk Percentage, HBP—Hit by Pitch, HR—Home Run, IsoP—Isolated Power, OBP—On-base Percentage, OPS—On-base Plus Slugging, PA—Plate Appearances, SF—Sacrifice Fly, SLG—Slugging Percentage, SO—Strikeout, SO%—Strikeout Percentage.

**Table 2 sports-13-00367-t002:** Study parameters obtained.

Characteristic	N = 14 ^1^
**Spatial Stroop task**	
Stroop RT (Horizontal) (ms)	23.5 (15.3)
Stroop RT (Vertical) (ms)	44.8 (27.6)
Stroop accuracy (Horizontal) (%)	95.1 (3.8)
Stroop accuracy (Vertical) (%)	96.2 (2.2)
**Ball selection task by VR**	
Ball selection accuracy (Total) (%)	67.8 (6.0)
Ball selection accuracy (ball) (%)	64.5 (9.7)
Ball selection accuracy (strike) (%)	71.6 (14.6)
**Batting statistics**	
BA	0.242 (0.080)
OBP	0.278 (0.089)
SLG	0.320 (0.122)
OPS	0.598 (0.204)
IsoP	0.078 (0.061)
BB/K	0.301 (0.248)
SO%	0.225 (0.099)
BB%	0.048 (0.032)

Note: ^1^ Mean (SD).

**Table 3 sports-13-00367-t003:** Correlation matrix of the Stroop task performance, VR batting selection task performance, and season batting performance parameters.

		Spatial Stroop Task				Ball Selection Accuracy by VR		
		RT (Horizontal)	RT (Vertical)	Accuracy (Horizontal)	Accuracy (Vertical)	Total	Ball	Strike
Ball selection accuracy by VR	Total	0.20	−0.48	0.05	−0.42			
	Ball	−0.37	0.36	−0.49	−0.08			
	Strike	0.45	−0.67 *	0.43	−0.30			
Batting performance	BA	0.25	−0.2	0.18	0.25	0.40	−0.09	0.46
	OBP	0.14	−0.31	0.21	0.17	0.57 *	−0.01	0.55 *
	SLG	0.27	−0.16	0.25	0.33	0.33	−0.22	0.51
	OPS	0.22	−0.23	0.24	0.27	0.45	−0.13	0.54 *
	IsoP	0.20	−0.06	0.25	0.32	0.14	−0.32	0.41
	BB/K	−0.15	−0.20	−0.10	0.00	0.45	0.10	0.32
	SO%	0.09	0.30	0.07	0.05	−0.31	−0.03	−0.25
	BB%	−0.20	−0.49	0.15	−0.22	0.82 *	0.22	0.58 *

The numbers in the matrix indicate the correlation coefficients between each variable, and the asterisks indicate significant correlations (*p* < 0.05).

**Table 4 sports-13-00367-t004:** Results of stepwise multiple regression analyses (only dependent variables for which a significant regression equation was obtained).

Dependent Variable	Independent Variable	Coefficient	Standard Error of Coefficient	R^2^	*p*-Value
OBP	Ball selection accuracy (Total)	0.009	0.004	0.322	0.034
OPS	Ball selection accuracy (Strike)	0.008	0.004	0.297	0.044
BB%	Ball selection accuracy (Total)	0.006	0.001	0.878	<0.001
	RT (Horizontal)	−0.001	0.000		
	Ball selection accuracy (Ball)	−0.001	0.000		

**Table 5 sports-13-00367-t005:** Comparison of mediation models.

Model	AIC	BIC	CFI	RMSEA
BB% Mediation	63.27	66.47	1.00	0.00
OBP Mediation	81.37	84.57	1.00	0.00
OPS Mediation	79.24	82.44	1.00	0.00

The BB% mediation model demonstrated the best fit with the lowest AIC (63.27) and BIC (66.47) values, suggesting that the relationship between executive function, pitch selection, and walk percentage was most strongly supported by our data.

**Table 6 sports-13-00367-t006:** Path coefficients for the BB% mediation model.

Path	Standardized Coefficient	Standard Error	*p*-Value	95% CI Lower	95% CI Upper
Stroop RT (Horizontal) → BB%	−0.380	0.147	0.010	−0.638	−0.085
Ball selection accuracy → BB%	0.897	0.097	<0.001	0.651	1.055
Stroop RT (Horizontal) → Ball selection accuracy	0.198	0.234	0.397	−0.350	0.508
Indirect effect	0.178	0.214	0.404	−0.308	0.464
Total effect	−0.202	0.232	0.384	−0.787	0.125

## Data Availability

The datasets generated and analyzed during the current study are not publicly available due to ethical restrictions imposed by the institutional review board to protect participant confidentiality, but are available from the corresponding author upon reasonable request.

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
