# Peer review of "Pitch Selection Ability and Spatial Executive Function Independently Predict Baseball Batting Performance"

_sports, 2025, doi:10.3390/sports13100367_

Round 1

Reviewer 1 Report

Comments and Suggestions for Authors

This manuscript investigates the relationship between pitch selection ability, cognitive function, and batting performance in collegiate baseball players. The authors employ a virtual reality-based pitch selection task and conduct correlational, regression, and mediation analyses. The study is novel and addresses an important question in sports science, offering practical implications for training. However, there is still space for improvement. I've included my comments on individual sections of the manuscript below.
Introduction
The introduction is well-structured and provides appropriate context for the research, identifying a gap in the literature regarding the quantification of pitch selection ability and its cognitive underpinnings. The discussion of pitch recognition and the cognitive demands of batting is well-supported with references. However, the use of the term “Batting Eye” is inconsistent and potentially confusing. It is not clearly defined and seems to refer variously to perceptual skill, pitch selection accuracy, and decision-making performance. This should be clarified, with a precise operational definition maintained throughout the manuscript. Additionally, the hypothesis near the end of the introduction is too general and would benefit from being more specific regarding which cognitive functions and which batting metrics were expected to correlate.
Materials and Methods
The use of a VR-based pitch selection task is particularly innovative and aligns well with the real-world perceptual demands of baseball. The description of how the videos were constructed and presented is clear. The Spatial Stroop task is a reasonable choice to assess executive function, although its scope is narrow and may not capture the full range of cognitive demands relevant to batting. One major limitation, acknowledged later in the discussion but not emphasized here, is the small sample size of only 14 participants. This substantially limits the statistical power of the correlation and mediation analyses.
Results
The authors report several significant associations between pitch selection accuracy (especially strike judgments) and batting metrics such as OBP, OPS, and BB%. These findings are interesting and potentially important. However, some of the language used to describe effect sizes (e.g., “strong correlation”) may be inappropriate given the sample size and the moderate magnitude of most correlations. The mediation analysis is clearly presented but requires more careful interpretation.
Discussion
The authors correctly note the significance of their findings in light of prior work, and their interpretation that pitch selection ability, as measured by their VR task, is meaningfully associated with batting outcomes is justified. However, the interpretation of the cognitive findings is more problematic. The role of executive function is both emphasized and dismissed in different parts of the discussion, which leads to a somewhat contradictory narrative. The claim that cognitive function “independently contributes” to batting performance should be tempered, as the data do not support a direct effect except in one limited case. Furthermore, the speculative connections made to neural mechanisms and dual-task performance, while interesting, extend beyond the scope of the study and should be more clearly framed as hypotheses for future research.

Author Response

Reviewer 1

Introduction

Comment1: The use of the term "Batting Eye" is inconsistent and potentially confusing. It is not clearly defined and seems to refer variously to perceptual skill, pitch selection accuracy, and decision-making performance. This should be clarified, with a precise operational definition maintained throughout the manuscript.

Response: Thank you for this important observation. We agree that the terminology requires clarification. We have decided to discontinue the use of the term “Batting Eye,” provide a clear operational definition of the term “Pitch selection” in the introduction, and consistently use this definition throughout the manuscript.

- Page 1, Lines 15-17:

  - Original: "The ability to determine whether a pitched ball has passed through the strike zone or the ball zone and to select a ball that can be hit is called "pitch selection," and may have a positive impact on batting average and other performance indicators."

  - Revised: " Pitch selection —the ability to discriminate balls from strikes— is fundamental to baseball batting success. This study examined whether this ability relates to executive function and batting performance in collegiate players."

- Throughout manuscript: Standardized terminology usage to align with this definition.

Comment2: The hypothesis near the end of the introduction is too general and would benefit from being more specific regarding which cognitive functions and which batting metrics were expected to correlate.

Response: We appreciate this suggestion and have revised our hypothesis to be more specific.

- Pages 1-2, Lines 84-86:

  - Original: "We hypothesized that cognitive functions would be positively related to both pitch discrimination ability and batting performance."

  - Revised: "We hypothesized that spatial interference processing, as measured by the Stroop task, would be positively related to pitch discrimination accuracy (particularly strike judgments), and that pitch selection ability would correlate with on-base percentage, walk percentage, and on-base plus slugging percentage."

Materials and Methods

Comment3: One major limitation, acknowledged later in the discussion but not emphasized here, is the small sample size of only 14 participants. This substantially limits the statistical power of the correlation and mediation analyses.

Response: We acknowledge this important limitation and have now addressed it explicitly in the Methods section.

Revision:

- Page 3, Lines 95-97:

  - Added: "A limitation of this study is the sample size of 14 participants, which may limit statistical power. Post hoc sensitivity analysis with 80% power and α=.05 demonstrated sufficient sensitivity to detect correlations exceeding r=0.63, computed using G*Power (3.1.9.2)."

Results

Comment4: Some of the language used to describe effect sizes (e.g., "strong correlation") may be inappropriate given the sample size and the moderate magnitude of most correlations.

Response: Thank you for this observation. We have revised our descriptions of effect sizes to be more conservative and appropriate.

Revision:

- Page 5, Line 180:

  - Original: "The walk percentage (BB%) also showed a strong significant correlation (r = 0.82, r = 0.58) with batting selection accuracy"

  - Revised: "The walk percentage (BB%) also showed a significant correlation (r = 0.82, r = 0.58) with batting selection accuracy"

Discussion

Comment5: The role of executive function is both emphasized and dismissed in different parts of the discussion, which leads to a somewhat contradictory narrative. The claim that cognitive function "independently contributes" to batting performance should be tempered, as the data do not support a direct effect except in one limited case.

Response: We appreciate this critique and have revised the discussion to present a more consistent and nuanced interpretation.

Revision:

- Page 8, Lines 246-249:

  - Original: "These results suggest that batting eye ability and cognitive function independently contribute to batting performance metrics related to pitch selection, rather than cognitive function influencing performance by enhancing pitch selection ability."

  - Revised: "Our mediation analysis revealed that batting eye ability and cognitive function appear to make separate, independent contributions to batting performance metrics, particularly walk percentage. However, we did not find evidence that cognitive function enhances batting performance by improving pitch selection ability, suggesting these are parallel rather than sequential processes."

- Page 9, Lines 303-305:

  - Added: "It is important to note that the direct effect of cognitive function on batting performance was limited to specific metrics, and our sample size constrains the generalizability of these findings."

Reviewer 2 Report

Comments and Suggestions for Authors

Manuscript: “Batting Eye, which supports batter performance in baseball players, is associated with executive function”

The study analyzed pitch selection in batting performance, a skill affecting various performance indicators. In general, I find the topic very interesting and useful in terms of research and sports practice. There are just a few suggestions that, in my opinion, could make the article stronger

Title:
In my opinion, the authors should modify the title to make it more concise and precise in relation to the study they propose, avoiding overlap with the keywords. E.g. Batting Eye and Attentional Inhibition of Baseball Players or E.g. Batting Eye and Spatial Attentional Control of Baseball Players, are just suggestions for the authors.

Objective and hypothesis: 
I suggest that the authors specify which executive function they are measuring, especially in the study hypothesis, considering that the only measure of executive functions is the spatial Stroop task. 

Experimental design: 
A statement on the experimental design used would be necessary in the methods section.

Tables: The tables are not always clear:
1) In the notes for Tables 2 and 3, add the abbreviation VR and its meaning.
2) Please, add two decimal places to the results in Table 5 (CFI and RMSEA columns)
3) Please adapt tables 5 and 6 to the MDPI template.

Results:
Lines 211- 233: Could the authors indicate the names of the variables (BB_per, OBP, OPS) in full the first time they appear, with the acronym in parentheses? This would make reading much easier.

Discussion: 
Authors should refer to the executive function they analyzed and not to cognitive functions or executive functions in general.

Formatting advices: 
I suggest that authors adapt the formatting of the title and bibliography to the MDPI template.

Author Response

Reviewer 2

Title

Comment1: The authors should modify the title to make it more concise and precise in relation to the study they propose, avoiding overlap with the keywords.

Response: Thank you for this suggestion. We have revised the title to be more concise and specific.

Revised Title: "Pitch Selection Ability and Spatial Executive Function Independently Predict Baseball Batting Performance"

Objective and Hypothesis

Comment2: I suggest that the authors specify which executive function they are measuring, especially in the study hypothesis, considering that the only measure of executive functions is the spatial Stroop task.

Response: We agree and have specified this in both the objectives and hypothesis.

Revision:

- Page 2, Lines 81-84:

  - Original: "Therefore, this study aimed to clarify whether pitch selection ability under conditions with minimal constraints on pitching judgment using virtual imagery relates to cognitive function and batting performance."

  - Revised: "Therefore, this study aimed to clarify whether pitch selection ability under conditions with minimal constraints on pitching judgment using virtual imagery relates to spatial interference processing (a specific executive function) and batting performance."

Experimental Design

Comment3: A statement on the experimental design used would be necessary in the methods section.

Response: Thank you for pointing this out. We have added this information.

Revision:

- Page 3, Lines 106-107:

  - Added: "This study employed a correlational design with mediation analysis to examine relationships between pitch selection ability, spatial interference processing, and batting performance metrics."

Tables

Comment4: In the notes for Tables 2 and 3, add the abbreviation VR and its meaning.

Response: Thank you for pointing that out. Since VR first appears on Page 3, Lines 97, we decided to spell it out in full there.

Revision:

- Page 3, Line 103:

- Original: “Fourteen college baseball players with batting records for the 2022-2023 season were subjected to the spatial Stroop task of the executive function task and the created VR pitch selection task.”

- Revised: “Fourteen college baseball players with batting records for the 2022-2023 season were subjected to the spatial Stroop task of the executive function task and the created Virtual Reality (VR) pitch selection task.”

Comment 5: Please, add two decimal places to the results in Table 5 (CFI and RMSEA columns)

Response: Done. All values in Table 5 now show two decimal places.

Revision:

- Table 5: CFI values now shown as "1.00" and RMSEA values as "0.00"

Comment 6: Please adapt tables 5 and 6 to the MDPI template.

Response: We have reformatted Tables 5 and 6 according to the Sports (MDPI) journal template. Additionally, Tables 1 through 4 have been reformatted to improve readability.

Results

Comment 7: Lines 211-233: Could the authors indicate the names of the variables (BB_per, OBP, OPS) in full the first time they appear?

Response: Regarding the variables you pointed out (BB%, OBP), we had already provided their full names on Page 5, Lines 179-180. However, we had not provided the full name for OPS, so we have corrected that section (Page 5, Line 183). For other variables as well, we have revised the wording to explicitly state their formal names at their first appearance in the text.

Discussion

Comment 8: Authors should refer to the executive function they analyzed and not to cognitive functions or executive functions in general.

Response: We agree and have revised the discussion accordingly.

Revision:

- Throughout Discussion: Changed "cognitive function" to "executive function" where appropriate.

Reviewer 3 Report

Comments and Suggestions for Authors

Please see the file attached

Author Response

Reviewer 3

Title

Comment1: Unclear, unconvincing, and does not highlight the specificity of the research. Possible misprint in the title, 'batter' instead of 'better'?

Response: Thank you for catching this. "Batter" is correct (referring to the baseball player), not "better." However, we agree the title needed clarification and have revised it as noted in response to Reviewer 2.

Revised Title: "Pitch Selection Ability and Spatial Executive Function Inde-pendently Predict Baseball Batting Performance"

Abstract

Comment 1: The abstract is unclear at the beginning, with a long and complex opening sentence. A more direct definition of pitch selection would improve comprehension.

Response: We have revised the opening of the abstract for clarity.

Revision:

- Abstract, Lines 15-17:

  - Original: "The ability to determine whether a pitched ball has passed through the strike zone or the ball zone and to select a ball that can be hit is called "pitch selection," and may have a positive impact on batting average and other performance indicators."

  - Revised: "Pitch selection—the ability to discriminate balls from strikes—is fundamental to baseball batting success. This study examined whether this ability relates to executive function and batting performance in collegiate players."

Comment 2: Essential details about the study sample are missing. It is crucial to specify the exact number of participants, their demographic characteristics, and their athletic level.

Response: We have added these details to the abstract. Additionally, we have added athletic level to Section 2.1.

- Abstract, Lines 20-22:

  - Original: "In this study, we created a pitch selection task in which participants watched pitching images captured by a 360-degree camera through virtual reality goggles to clarify whether pitch selection was related to batting performance and to cognitive function."

  - Revised: "14 male collegiate baseball players (age: 20.6 ± 1.0 years, first division university league) completed a virtual reality pitch selection task and spatial Stroop task."

-Page 3, Lines 90-92:

  - Original: "The participants were 14 players in the 2023 season who belonged to the hard ball team of [blinded for review], "

  - Revised: "The participants were 14 male players belonging to the Division I university baseball league,"

Comment 3: The methodological tools used should be explicitly stated.

Response: Now specified in the abstract.

- Abstract, Lines 20-22:

  - Added: "Methods included virtual reality pitch selection assessment, spatial Stroop task, and official batting statistics from league play."

Introduction

Comment 4: The introduction requires a logical reorganization with clear sections on: pitch selection concept, technical details, executive functions, research gap, and hypotheses.

Response: Thank you for this structural suggestion. We have reorganized the introduction accordingly.

Materials and Methods

Comment 5: The description of experimental procedures needs to be more specific, including laboratory environment, participant preparation, duration and order of tasks.

Response: We have added these details.

- Page 3, Lines 104-106:

  - Original: "After entering a quiet laboratory, the participants checked their physical condition and performed an executive functioning task, followed by a VR pitch selection task."

  - Revised: "Testing occurred in a quiet laboratory environment. Participants were instructed to avoid caffeine and vigorous exercise for 2 hours before testing. After a 5-minute rest period, participants completed the spatial Stroop task (approximately 10 minutes), followed by a 5-minute break, then the VR pitch selection task (approximately 20 minutes). Total session duration was approximately 40 minutes."

Discussion

Comment 6: The discussion should place greater emphasis on the existing gap in the literature and strengthen connections with previous studies.

Response: We have expanded the discussion to better address the literature gap and contextualize our findings.

- Page 8, Lines 254-257:

  - Added: "This study addresses a critical gap in understanding the independent contributions of perceptual-motor skills and cognitive processes to batting performance. Unlike previous work that found no relationship between general cognitive tasks and batting [8], our use of sport-specific assessment revealed significant associations."

- Page 8, Lines 240-243:

  - Added: "Our findings extend the work of Themanson et al. [7] by demonstrating these relationships under more ecologically valid conditions using VR technology."

Round 2

Reviewer 3 Report

Comments and Suggestions for Authors

The requested changes and additions have been made. I believe that the paper can be accepted in its current form.